# A Preliminary Investigation of Mobile Respiratory Function Testing in Western Australian Communities

Petra Czarniak [1,*], Kim Watkins [1], Finbarr Foy [1], Richard Parsons [1], Graham L. Hall [2] and Bruce Sunderland [1]

1   Curtin Medical School, Faculty of Health Sciences, Curtin University, Bentley, WA 6102, Australia
2   School of Allied Health, Faculty of Health Sciences, Curtin University, Bentley, WA 6102, Australia
*   Correspondence: p.czarniak@curtin.edu.au

**Abstract:** Although underutilized, spirometry is essential in the diagnosis and management of chronic obstructive pulmonary disease (COPD) and asthma. This study aimed to investigate a mobile (i.e., transportable) lung function testing (LFT) services in two metropolitan and two rural clinics in Western Australia. Individuals attending a mobile LFT clinic in 2021 were invited to complete questionnaires at baseline and after 6–8 weeks. Questionnaires were completed by 59/74 (79.7%) respondents (mean age 62.5 ± 14.2 years); most were female (35/59; 59.3%). A history of asthma was reported in 50.9% (30/59) and COPD in 18.6% (11/59) of respondents [13.6% (8/59) reported both]. At baseline, most (22/30; 73.3%) had asthma control test scores ≤19 (mean 16.6; range 8.0–25.0); at follow-up, 16/31 (51.6%) had scores ≤19 (mean score 18.0; range 6.0–25.0). Of the 11 diagnosed with COPD at baseline, the mean Clinical COPD Questionnaire and COPD assessment test scores were greater at follow-up (1.9 vs. 2.3; and: 10.3 vs. 14.7 respectively), reflecting worsening disease. Most participants (57/59; 96.6%) were satisfied with the LFT experience. The role of mobile LFT services to optimize the diagnosis and management of chronic lung disease and to minimize patient burden requires further investigation to improve short term patient outcomes.

**Keywords:** asthma; chronic obstructive pulmonary disease; COPD; spirometry; lung function testing

## 1. Introduction

Chronic obstructive pulmonary disease (COPD) and asthma are prevalent chronic lung diseases associated with significant morbidity and mortality [1,2]. National statistics in Australia in 2020 reported 417 deaths due to asthma (many of which were preventable) and listed COPD as the fifth leading cause of death [3–5]. One of the challenges in optimal management of these chronic respiratory conditions is the lack of accessibility to health services, especially for patients living in rural areas [6]. In Western Australia (WA), there is evidence that rural residents are more likely than metropolitan residents to have potentially preventable hospitalisations for respiratory disease [7]. Correct diagnosis of asthma or COPD and appropriate patient education/supported self-management are vital. An essential component for the diagnosis and appropriate management of COPD and asthma is spirometry, which should be performed by suitably trained staff [8]. Incorrect diagnosis, which can lead to mismanagement of patient lung health, has been reported in 12–50% of cases [9]. Spirometry should be undertaken at diagnosis and every 1 to 2 years as part of disease monitoring and review, but there is significant evidence that it is an underutilised test [6,10,11]. There is also evidence that the frequency of spirometry testing may be significantly lower in rural centres compared to metropolitan centres [6].

In WA, there are several options to access lung function testing. A full spectrum of lung function tests is available at tertiary laboratories located in hospitals in the metropolitan area. However, these may be difficult for rural residents to access. Although many general practitioner (GP) surgeries have spirometers, use of spirometry testing as a clinical tool is low [9,10]. Alternative testing is performed via private respiratory laboratories, some of

which provide a mobile (i.e., transportable) lung function testing service. Private laboratory testing potentially provides advantages over hospital or GP-based testing in terms of accessibility and quality of service. Mobile services are accessible in both metropolitan and rural areas of WA, offer a broad range of respiratory function tests, are generally conducted by qualified respiratory scientists with results reviewed and reported by respiratory physicians before being made available to the primary care physician to guide the patients' clinical management.

The aim of this pilot study was to investigate the potential benefits and overall impact of a mobile respiratory function testing service in metropolitan and rural areas of WA. Specific objectives included discerning patient disease management before and after participating in the mobile lung function testing service and investigating GP perceptions of the benefit of the mobile lung function testing service.

## 2. Methodology

### 2.1. Ethical Approval

This study was approved by the Curtin University Human Research Ethics Committee (HREC) (HRE2020-0130) on the 19 March 2020.

### 2.2. Setting

This convenience sample, prospective pilot study recruited individual participants 18 years or older, who attended a mobile lung function clinic in WA at four different sites—two rural sites (Busselton and Narrogin) and two metropolitan sites (Carine and Rockingham) (approximately 15–20 from each site). These sites were chosen to optimise recruitment numbers, as historically, they had been well-attended by patients. Each person attending a testing clinic did so for a reason (which they explained). They were invited to participate in the study after they were provided with a study 'Participant Information Sheet', 'Participant Consent Form' and advised that the questionnaires would take 20–30 min to complete. In addition, they were also asked to provide consent for the researchers to contact their GP. Individuals aged <18 years were excluded.

Where consent was provided, a letter was sent to the GPs, explaining that one of their patients (name not specified) agreed to participate in a mobile lung function-testing study. The GP was provided with written information about the study and invited to access an electronic link to complete a short questionnaire (10–15 min) on the REDCap® platform.

### 2.3. Study Design

Two online questionnaires, administered via the RedCap platform, were used to investigate the potential benefits and impact on patients who presented for lung function testing at a mobile service. Lung function testing was conducted by an experienced respiratory scientist employed by the mobile lung function testing clinic and involved various tests including spirometry, lung volume and diffusion capacity testing. Lung function testing results were analysed by a respiratory physician associated with the mobile lung function testing clinic and a report was forwarded to the patient's GP.

The two questionnaires, a baseline questionnaire and a 6-8 week follow-up questionnaire, were developed using a mixed methods format including a combination of previously validated tools along with demographic and medical history questions. Validated tools incorporated into the questionnaires included the asthma control test (ACT) [10], COPD assessment test (CAT) [11], the clinical COPD questionnaire (CCQ) [12] and the St George's respiratory questionnaire (SGRQ) [13] Permission was sought and granted to use these validated tests.

Briefly, the ACT contained five items with a four week recall on symptoms and daily functioning. Each item was scored from 1 to 5. The ACT overall scores ranged from 5 (poorly controlled asthma) to 25 (completely controlled asthma), with an ACT >19 indicating well-controlled asthma. The CAT, which assessed the impact of COPD on current health status, consisted of eight items, each scored between 0 and 5. Scoring involved

adding the scores of the eight individual items to provide a total score out of 40. Higher scores represented a greater negative impact on disease. The CCQ consisted of 10 items, and all scores ranged from zero to six (0 = no impairment, 6 = worst score). The three domains of the CCQ included symptoms (4 items), functional state (4 items) and mental state (2 items). The recall period was seven days. The SGRQ, an airway disease-specific questionnaire to measure the impact on overall health, consisted of 50 items and three domains. Domains included symptoms (8 items), activity (16 items) and impacts (26 items). Each score ranged from 0 to 100% (with higher scores indicating more limitations).

One of the investigators recruited patients and collected data at each lung function testing site. They assisted with the administration of the baseline questionnaire's completion using an iPad. An automated follow-up email was sent to participants after 6–8 weeks via REDCap®, inviting them to complete the follow-up questionnaire. Participants unable to access the questionnaire electronically were invited to complete the same questionnaire as a telephone interview. Non-responders were telephoned two weeks after the link to the follow-up questionnaire becoming available to them. Participants who completed the baseline and follow-up questionnaire were offered an AUD 50 gift card as a token of appreciation of their time.

The GP questionnaire collected demographic data (age, gender, years of general practice, etc.), and asked GPs about their awareness, and usefulness, of the lung function testing service and respiratory physician's subsequent report. GPs were also asked to provide information about any actions/intentions in response to the respiratory physician's report/s following testing. The study aimed to recruit 20 GPs. GPs who agreed to participate were offered an AUD 50 gift card as a token of appreciation for their time.

### 2.3.1. Baseline Questionnaire

The baseline questionnaire was divided into five sections and consisted of 54 questions. The sections were Part A: participant demographic details (14 questions); Part B: past medical and pulmonary history of participant (8 questions); Part C: questions for participants with an asthma diagnosis (9 questions that included the ACT); Part D: questions for participants with COPD (8 questions that included the CCQ and CAT); Part E: quality of life assessment for patients with a lung condition (only SGRQ) (15 questions). Questions in Part C and D were only accessible to respondents who, in Part B, indicated they had asthma or COPD.

### 2.3.2. Follow-Up Questionnaire

The follow-up questionnaire was designed to evaluate participant's experience and satisfaction with the lung function testing service and investigate the impact of lung function testing (subsequent GP appointment, confirmation of a respiratory diagnosis and therapeutic changes). The questionnaire was divided into five sections and consisted of 33 questions. The sections were Part A: experience and satisfaction with the lung function testing by the mobile respiratory service (14 questions); Part B: respiratory medications (1 question); Part C: questions for participants with a diagnosis of COPD (2 questions consisting of the CCQ and CAT); Part D: questions for participants with a diagnosis of asthma (1 question consisting of the ACT); Part E: quality of life assessment for patients with a lung condition (SGRQ) (15 questions). Questions in Part C and D were only accessible to respondents diagnosed with asthma or COPD. Questions relating to the experience and satisfaction with lung function testing by the mobile respiratory service included five-point Likert scale statements (from 1 = strongly agree to 5 = strongly disagree), as well as yes/no responses or responses involving selecting from several options.

### 2.3.3. General Practitioner Questionnaire

The GP questionnaire was divided into three sections and consisted of 20 questions. The sections were Part A: GP details (13 questions); Part B: awareness and usefulness of the lung function test and report (5 questions); Part C: actions taken in response to report (2 questions). In Part B, one of the questions asked GPs if they were aware that

one of their patients had recently attended a mobile lung function testing service provided by a respiratory testing service. Respondents who replied 'yes' were then asked their level of agreement with several statements about the mobile lung function testing service and respiratory physician's report, for their patient/patients. The level of agreement to statements was based on a five-point Likert scale (from 1 = strongly agree to 5 = strongly disagree). Information gathered in Part C related to GP's actions/intentions in response to the lung function testing report/s using the following response options: 'have actioned', 'intend to action', 'do not intend to action' and 'not applicable to diagnosis'. The final question was open-ended and asked respondents to specify any other actions taken for each patient that was attributed to receiving the lung function testing report.

All questionnaires were assessed by academics and non-academics (who did or did not have lung disease), for face validity.

## 3. Analysis

All data, including data from the participant baseline questionnaire and 6–8 week follow-up questionnaire, were de-identified and analysed using SAS version 9.4 (SAS Institute Inc., Cary, NC, USA. 2016). Simple descriptive statistics (frequencies and percentages for categorical variables, means, standard deviation and medians for variables measured on a continuous scale) were used to summarise patient demographic data. Student's *t*-test was used to compare test scores between rural and metropolitan clinics. Differences in various validated test scores at baseline and follow-up between metropolitan and rural areas were determined. *p*-values were obtained from two *t*-tests (*t*-test and *t*-test Satterthwaite), comparing the various scores between metro and rural, as these cohorts may not have the same variances. Furthermore, the F-test was used to indicate whether the *p*-values from the *t*-test and *t*-test Satterthwaite should be used, based on whether the F-test was significant ($p < 0.05$). Where this occurred, the Satterthwaite *p*-value was reported. For all other results, the standard *t*-test *p*-values were reported. The a priori level of significance for all statistical tests was set at $p < 0.05$.

## 4. Results

Of 74 respondents who consented to participate in the study, 59 (79.7%) completed the baseline and follow-up questionnaires. Table 1 provides a summary of demographic data. Most respondents were recruited from Rockingham (33.9%) and Busselton (27.1%) clinics, and most were female (35/59; 59.3%). The mean age was 62.5 ± 14.2 years (range from 27–84 years). The majority of respondents (36/59; 61.0%) had worked in smoky or dusty environments such as farms, mining or industrial environments, cleaning and painting (Table 2). Thirty respondents (50.9%) reported a history of asthma and 11 a history of COPD (18.6%). Of these respondents, 8 (13.6%) reported both asthma and COPD.

Approximately half of the respondents had never smoked (29/59; 49.1%), with 24 (40.7%) identifying as former smokers and 6 (10.2%) as current smokers. A significant proposal of respondents had multiple comorbidities (45/59; 76.3%), the most common being asthma (30/59; 50.9%) and back pain (23/59; 39.0%). Asthma diagnosis was most commonly made by a doctor (24/30; 80.0%).

Most respondents had had a previous lung function test (32/59; 54.2%) (Table 1), which was most commonly performed by a respiratory scientist (14/30; 46.7%). The most common reason for the current lung function test was that their GP had referred them (31/59; 52.5%) or they were referred by a respiratory specialist (29/59; 49.2%) (one person selected more than one response).

Most respondents (48/59; 81.4%) did not use any respiratory medicines. For patients using pharmacotherapy, commonly used respiratory medications (>5) included the beta$_2$-agonist salbutamol, corticosteroids (ciclesonide) and a combination of tiotropium/olodaterol.

**Table 1.** Baseline data of participants who had a lung function test.

| Parameter | | n | % |
|---|---|---|---|
| Clinic (*n* = 59) | Rockingham | 20 | 33.9 |
| | Carine | 12 | 20.3 |
| | Narrogin | 11 | 18.6 |
| | Busselton | 16 | 27.1 |
| Gender (*n* = 59) | Male | 24 | 40.7 |
| | Female | 35 | 59.3 |
| Age (*n* = 59) | 20–29 | 1 | 1.7 |
| | 30–39 | 5 | 8.5 |
| | 40–49 | 4 | 6.8 |
| | 50–59 | 11 | 18.6 |
| | 60–69 | 18 | 30.5 |
| | 70–79 | 14 | 23.7 |
| | 80–89 | 6 | 10.2 |
| Country of birth (*n* = 59) | Australia | 37 | 62.7 |
| | Canada | 1 | 1.7 |
| | England | 1 | 1.7 |
| | Italy | 1 | 1.7 |
| | New Zealand | 4 | 6.8 |
| | Scotland | 3 | 5.1 |
| | Singapore | 1 | 1.7 |
| | South Africa | 1 | 1.7 |
| | USA | 2 | 3.4 |
| | United Kingdom | 8 | 13.6 |
| Aboriginal and Torres Strait Islander (ATSI) (*n* = 59) | Yes | 7 | 11.9 |
| | No | 52 | 88.1 |
| Employment (*n* = 59) | Self-employed | 10 | 16.9 |
| | Employed for wages | 10 | 16.9 |
| | Homemaker | 5 | 8.5 |
| | Retired | 29 | 49.2 |
| | Not employed/looking for work | 1 | 1.7 |
| | Not employed/not looking for work | 2 | 3.4 |
| | Student | 1 | 1.7 |
| | Unable to work | 1 | 1.7 |
| Smoking status (*n* = 59) | Never smoked | 29 | 49.1 |
| | Former smoker | 24 | 40.7 |
| | Current smoker | 6 | 10.2 |
| Reasons for lung function test (participants could choose more than one response) | Confirmation of diagnosis | 9 | 15.2 |
| | Uncontrolled symptoms/exacerbation of disease/not feeling good | 8 | 13.6 |
| | The specialist requested the test (respiratory specialist referral) | 29 | 49.2 |
| | General practitioner referred me (GP referral) | 31 | 52.5 |
| | A friend suggested I have a lung function test | 0 | 0.0 |
| | I decided to attend (self-referral) | 1 | 1.7 |
| | I saw an advertisement for the lung function testing | 0 | 0.0 |
| | I was curious | 0 | 0.0 |
| | I am looking for answers to improve my health | 2 | 3.4 |
| Previous lung function test | No | 27 | 45.8 |
| | Yes | 32 | 54.2 |
| How long ago was the previous lung function test? (*n* = 32) | <12 months | 19 | 59.4 |
| | 1–3 years | 8 | 25.0 |
| | >5–10 years | 1 | 3.1 |
| | >10 years | 4 | 12.5 |

**Table 1.** *Cont.*

| Parameter | | *n* | *%* |
|---|---|---|---|
| Most common medical conditions (i.e., medical conditions reported by > 10 participants) | Anxiety | 14 | 23.7 |
| | Asthma | 30 | 50.9 |
| | Back pain | 23 | 39.0 |
| | COPD | 11 | 18.6 |
| | Hay fever | 18 | 30.5 |
| | Hypertension | 19 | 32.2 |
| | High cholesterol | 11 | 18.6 |
| Respiratory medications used in the past week (*n* = 59) | None | 48 | 81.4 |
| | Some | 11 | 18.6 |
| Who made the diagnosis of asthma (*n* = 30) | My doctor | 24 | 80.8 |
| | A nurse | 0 | 0.0 |
| | Medical specialist | 2 | 6.7 |
| | During a hospital admission | 1 | 3.3 |
| | Friend | 0 | 0.0 |
| | Other (please specify) | 3 | 10.0 |
| Asthma Action Plan (*n* = 30) | Yes | 8 | 73.3 |
| | No | 22 | 26.7 |
| Asthma Control Test Score (*n* = 30) | ≤19 | 22 | 73.3 |
| | >19 | 8 | 26.7 |
| Diagnosis of COPD (some respondents chose more than one response) | My doctor | 9 | * |
| | A nurse | 0 | * |
| | Medical specialist | 4 | * |
| | During a hospital admission | 1 | * |
| | Friend | 0 | * |
| | Other (please specify) | 0 | * |

* As respondents chose more than one response, a percentage was not calculated.

**Table 2.** Type of work-related activities that exposed respondents to a smoky or dusty environment.

| Work Related Activities That Exposed Respondents to a Smoky or Dusty Environment (*n* = 36) | | |
|---|---|---|
| Barmaid | Dust-filled environment | Navy, ship supplies |
| Bars/clubs | Electrician | Oil and gas chemical plant |
| Born on a farm | Farm | Painter |
| Born on a farm—involved with chickens | Farm work/road construction | Painting |
| Building industry | Furniture making | Sawmill; working around smokers |
| Cargo | Hospital | Sewage worker |
| Carpet layer | Industrial environment | Shearing |
| Cleaner | Lived on a farm | Smoking in staff rooms |
| Cleaner at mine sites | Lived with a smoker | The royal show |
| Cleaning | Mining | Welding/boilermaker |
| Coal miner | Mowing, spraying, road works | |

### 4.1. Satisfaction

Most respondents (57/59; 96.6%) were satisfied with the lung function testing experience (Figure 1) and would recommend the service to others.

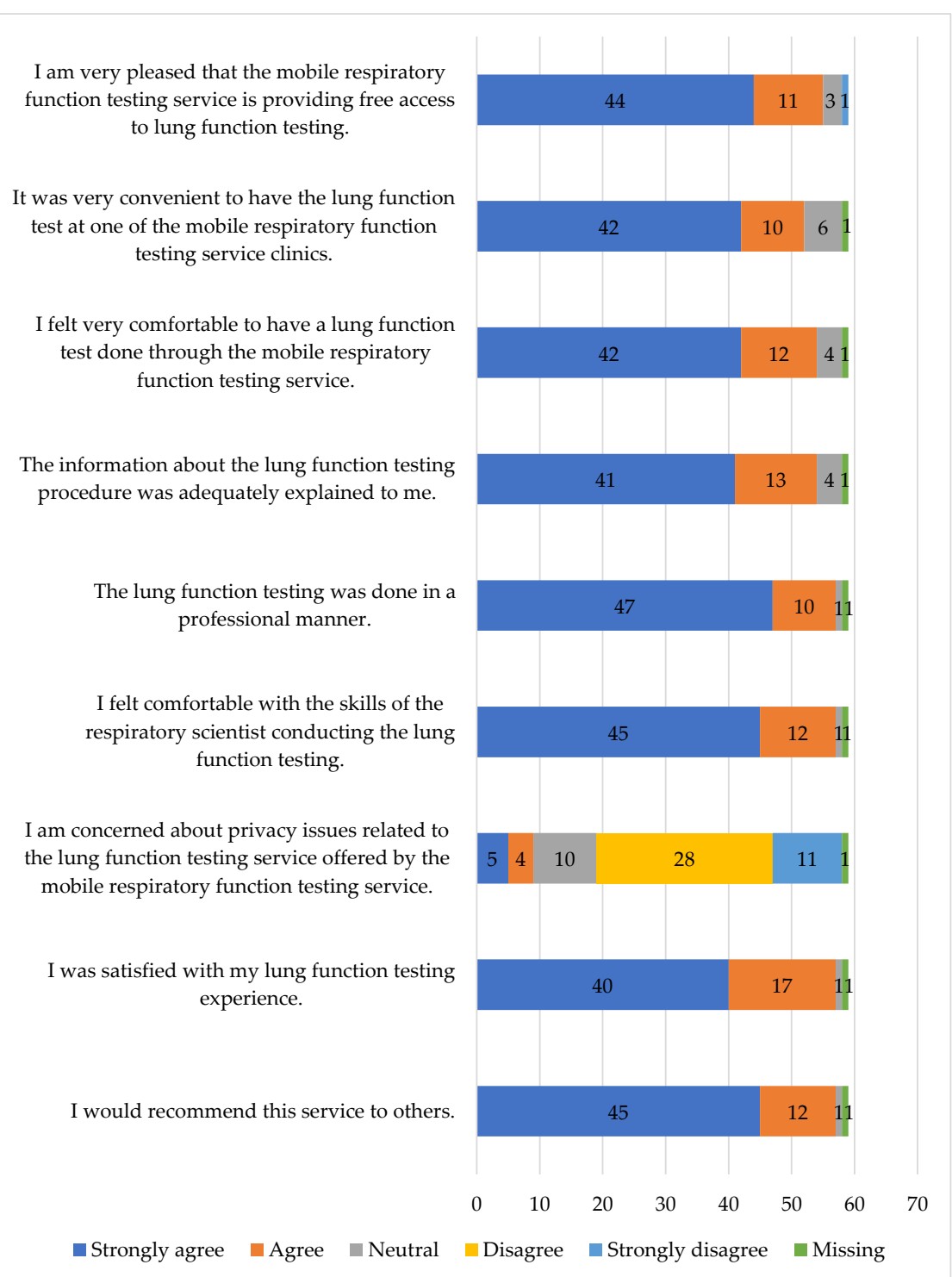

**Figure 1.** Participants' satisfaction with the mobile lung function testing service.

### 4.2. Respondents with Asthma or COPD

All 30 respondents who had been previously diagnosed with asthma, reported that they had experienced symptoms of asthma, such as wheeze, chest tightness, coughing or shortness of breath, in the past 12 months (Figure 2). Most respondents (21/29; 72.4%) visited their GP at least once in the past 12 months due to worsening or out of control asthma symptoms, while only eight (27.6%) did not consult their GP. Most (24/30; 80.0%) did not have to go to hospital or the emergency department in the past 12 months due

to worsening or out of control asthma symptoms, although four respondents (13.3%) had a single hospital presentation in the past 12 months, one respondent (3.3%) had two presentations, and another respondent (3.3%), who lived remotely in Narrogin, had 12 hospital presentations in the past 12 months for these reasons.

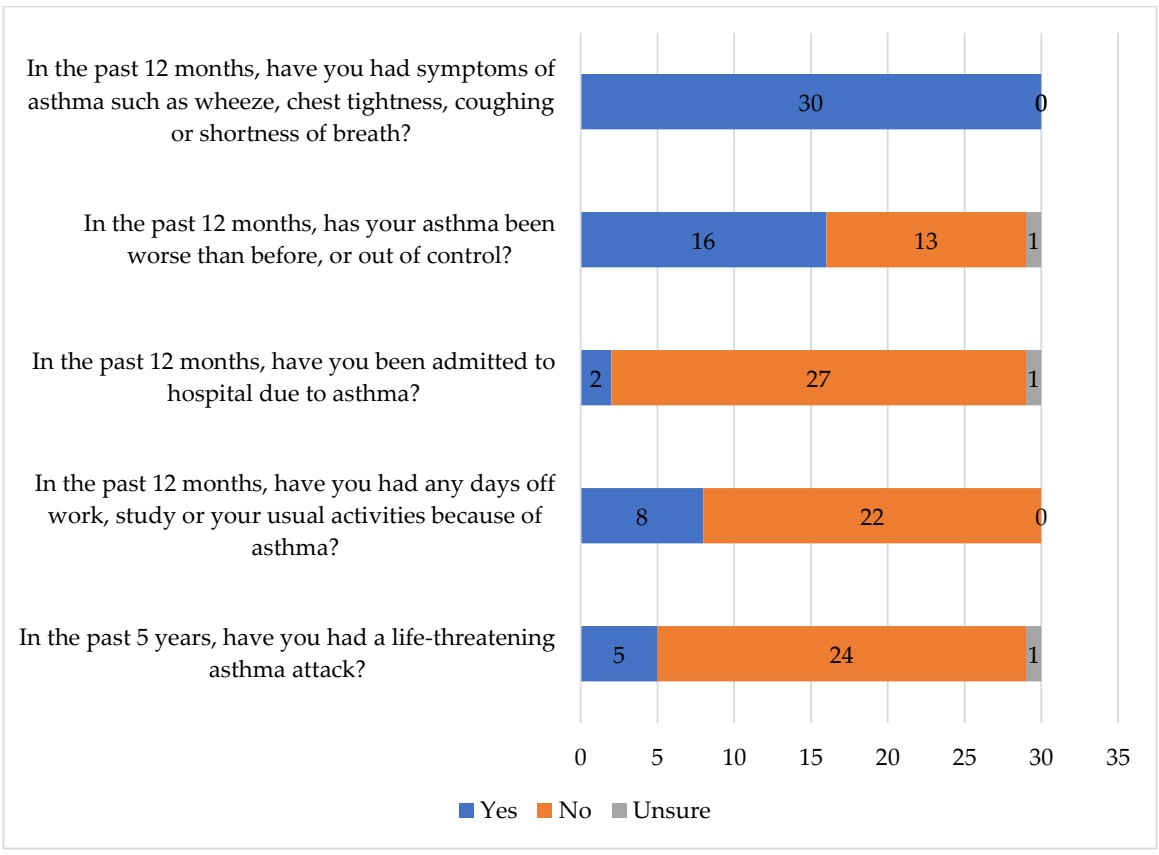

**Figure 2.** Respondents' agreement with statements related to asthma.

Only eight (26.7%) respondents with asthma had an Asthma Action Plan and three had referred to it in the past 12 months to manage their asthma. Most (7/8; 87.5%) felt confident to use their Asthma Action Plan.

At baseline, most respondents (22/30; 73.3%) had ACT scores $\leq$ 19, indicating their asthma was poorly controlled (mean 16.6; range from 8–25) (Table 3). Although there was some improvement in the ACT score at follow-up (mean 18.0; range 6.0–25.0), 16/31 respondents had ACT scores $\leq$ 19. In comparing each variable from baseline to follow-up, while there were statistically significant changes for some variables, the sample size for most variables were small (Table 3). Interestingly, not all respondents who identified having asthma in the baseline questionnaire were diagnosed with asthma after the lung function test (5/30; 16.7%). Following the lung function test, 6/31 (19.4%) respondents, who did not indicate they had asthma at baseline, reported they had asthma, two of whom had an ACT score < 19.

Of the 11 respondents who reported having been diagnosed with COPD at baseline, five (45.4%) had experienced COPD exacerbations in the past 12 months (Figure 3). In addition, two respondents reported that the exacerbations impacted their usual activities and that they had been admitted to hospital (one of the respondents had experienced both, i.e., the exacerbations impacted their usual activities and that they had been admitted to hospital). Following the lung function test, a total of 13 respondents indicated they had COPD; 2 respondents, who did not report having COPD at baseline, indicated that they were diagnosed with COPD. Most (8/11; 72.7%) were unaware of the severity of their COPD.

**Table 3.** Comparison of validated test scores at baseline and follow-up. *p*-values were obtained from the paired *t*-test to compare the change in means from baseline to follow-up.

| Variable | Baseline | | | Follow-Up | | |
|---|---|---|---|---|---|---|
| | Mean | Median | Range | Mean | Median | Range |
| ACT Score (*n* = 30; *n* = 31; *p* = 0.154) | 16.6 | 17.0 | 8.0–25.0 | 18.0 | 19.0 | 6.0–25.0 |
| CCQ Total Score (*n* = 11; *n* = 13; *p* = 0.068) | 1.9 | 2.1 | 0.1–3.8 | 2.5 | 2.3 | 0.9–3.9 |
| CCQ Symptom Score (*n* = 11; *n* = 13; *p* = 0.035) | 2.2 | 2.0 | 0.2–4.5 | 3.0 | 2.8 | 1.3–4.8 |
| CCQ Functional Score (*n* = 11; *n* = 13; *p* = 0.506) | 1.6 | 1.5 | 0.0–4.0 | 2.0 | 2.0 | 0.5–3.8 |
| CCQ Mental Score (*n* = 11; *n* = 13; *p* = 0.382) | 1.8 | 1.5 | 0.0–5.5 | 2.5 | 2.5 | 0.5–4.0 |
| CAT score (*n* = 11; *n* = 11; *p* = 0.034) | 10.3 | 10.0 | 0.0–26.0 | 14.7 | 16.0 | 7.0–27.0 |
| St George Score (*n* = 43; *n* = 45; *p* = 0.851) | 36.2 | 37.3 | 0.9–69.2 | 33.6 | 35.2 | 2.0–67.3 |
| St George Symptom Score (*n* = 43; *n* = 49; *p* = 0.220) | 50.8 | 53.0 | 7.5–90.0 | 43.3 | 40.9 | 0.0–97.3 |
| St George Activity Score (*n* = 59; *n* = 54; *p* = 0.0015) | 48.6 | 53.5 | 0.0–87.3 | 53.2 | 57.7 | 0.0–92.5 |
| St George Impacts Score (*n* = 59; *n* = 57; *p* = 0.045) | 20.7 | 18.9 | 0.0–58.7 | 17.7 | 15.5 | 0.0–50.3 |

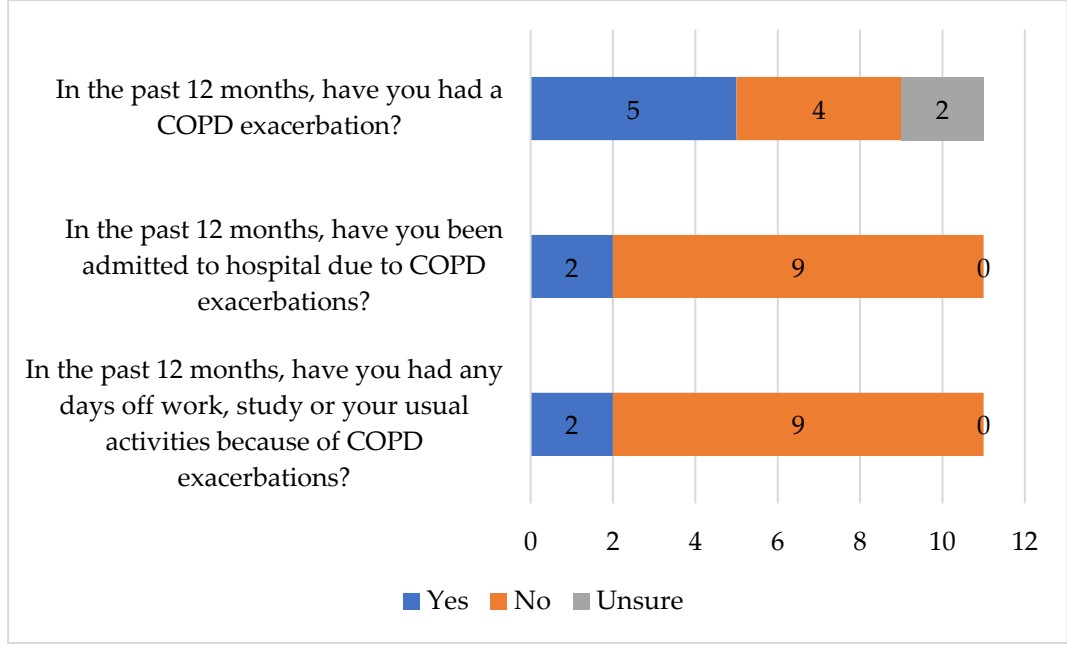

**Figure 3.** Respondents' agreement with statements related to COPD.

Compared to baseline results, the mean overall St George Score, the Symptom Score and Impact Score decreased at follow-up. However, the mean St George Activity Score increased at follow-up, suggesting more limitations of activity (Table 3).

The summary scores shown in Table 4 were compared between participants attending the metropolitan and rural clinics, and there appeared to be no significant differences.

**Table 4.** Differences in various validated test scores at baseline and follow-up between metropolitan and rural areas.

| Variable and Baseline, Follow-Up and Change Score | | Metropolitan (Mean) | Rural (Mean) | *p* |
|---|---|---|---|---|
| ACT score | Baseline | 16.12 | 17.23 | 0.517 |
| | Follow-up | 16.88 | 19.43 | 0.160 |
| | Change | 0.43 | 2.55 | 0.264 |
| CCQ total score | Baseline | 2.23 | 1.46 | 0.240 |
| | Follow-up | 2.61 | 2.30 | 0.401 * |
| | Change | 0.40 | 0.53 | 0.796 |
| CCQ symptom score | Baseline | 2.83 | 1.40 | 0.067 |
| | Follow-up | 3.09 | 2.75 | 0.561 |
| | Change | 0.46 | 1.13 | 0.289 |
| CCQ functional score | Baseline | 1.92 | 1.25 | 0.446 |
| | Follow-up | 2.22 | 1.70 | 0.308 |
| | Change | 0.38 | 0.06 | 0.697 |
| CCQ mental score | Baseline | 1.67 | 2.00 | 0.775 |
| | Follow-up | 2.44 | 2.60 | 0.795 |
| | Change | 0.33 | 0.25 | 0.909 |
| CAT Sscore | Baseline | 12.33 | 7.80 | 0.298 |
| | Follow-up | 16.67 | 12.40 | 0.261 |
| | Change | 5.25 | 1.50 | 0.156 |
| St George score | Baseline | 38.88 | 33.16 | 0.292 |
| | Follow-up | 36.37 | 30.10 | 0.246 |
| | Change | 0.82 | −1.66 | 0.405 |
| St George symptom score | Baseline | 52.46 | 48.97 | 0.641 |
| | Follow-up | 45.56 | 40.07 | 0.521 |
| | Change | −0.63 | −8.84 | 0.207 |
| St George activity score | Baseline | 51.53 | 45.05 | 0.324 |
| | Follow-up | 56.32 | 49.82 | 0.399 |
| | Change | 5.37 | 4.86 | 0.857 |
| St George impacts score | Baseline | 22.78 | 18.25 | 0.272 |
| | Follow-up | 19.62 | 15.41 | 0.268 |
| | Change | −2.66 | −2.14 | 0.828 |

* Satterthwaite *p*-value.

### 4.3. General Practitioner Questionnaire

Nine GPs consented to participate in the study but only six completed the questionnaire. All GPs held a spirometer in their surgery; three never used it or had not used it in the past 6 months, while two used it several times a week. Only two GPs were aware of the mobile lung function testing clinic. One GP had referred a patient and received a lung function test report. The GP reported that the information in the report was concise and easy to read, and that the report, and the results provided, were helpful for patient diagnosis or patient management. Five GPs advised they did not receive a report.

## 5. Discussion

The aim of this pilot study was to investigate the potential benefits and overall impact of a mobile respiratory function testing service in metropolitan and rural areas of WA. We found that the majority of participants were satisfied with the lung function-testing experience and would recommend the service to others. However, most GPs were not aware of the mobile lung function testing clinic.

We found that most participants had poorly controlled asthma indicated by an ACT score ≤19. Despite this, six respondents who had an ACT score ≤19 perceived their asthma to be well-controlled. Similar findings were reported in a UK study where only 32.1% of patients with ACT scores ≤19 considered their asthma was uncontrolled [12]. A finding in this study was that the ACT score did not improve significantly after follow-up, when it was expected participants would have consulted their GP to optimise therapy. Notably eight weeks after the lung function test, some participants had not consulted their GP, which may have been due to a lack of appointment availability. However, given the lack of GP awareness of the mobile lung function testing clinic found in this pilot study, and not knowing when the respiratory physician's report was sent to the GP, there is very little chance that the service can improve disease outcomes.

There is evidence that some participants may tolerate symptoms indicative of poor asthma control as part of living with asthma [13]. In a qualitative study in the UK, researchers reported that some study participants had an 'internal barometer', which set 'out of control' asthma symptoms considerably higher than the ACT score. These participants also placed less importance on asthma medications for their current state of health [13].

It should be noted that where participants had consulted their GP and changes were initiated in response to lung function testing, there could be a lag time for achievement of optimum symptom control (for corticosteroids, this can be several weeks). As optimum treatment also requires advice on avoiding triggers, patient education around this is an important component, but for some people, especially those living in rural areas on farms, or those exposed to chemicals at work, this may be challenging. An initial step in optimising treatment for lung disease is to obtain a correct diagnosis and for those with poorly managed lung disease; therapy should be reviewed, and symptoms monitored.

Of those with asthma, very few had an Asthma Action Plan, despite most participants experiencing respiratory symptoms in the previous 12 months. Provision of an Asthma Action Plan and individualised asthma education has been shown to reduce asthma severity scores, unscheduled consultations, hospitalisations and asthma-specific quality of life [14,15]. Furthermore, international guidelines strongly recommend that all patients with asthma receive a written Asthma Action Plan [16]. GPs should be educated on the importance of Asthma Action Plans to empower their patients to achieve better control of asthma symptoms and reduce hospitalisations.

Although based on small numbers, some of the disease state characteristics of respondents based on the St George Activity Score and Impacts Score, indicated a significant worsening of symptoms at follow-up. For most participants with COPD, validated test results, such as the CCQ symptom score and CAT score, also showed a greater negative impact of disease at follow-up compared to baseline. This was an unexpected finding that may have been due to seasonal variation. Alternatively, for those with newly diagnosed COPD, perhaps the reality of having this condition influenced their behaviour (i.e., illness anxiety disorder). For many of these participants, it would have been valuable to be able to refer themselves or be referred for lung function testing to obtain an insight into their condition. In the long term, this should lead to optimization of disease management.

This study provided two questionnaires to participants (baseline and follow-up) to investigate patient disease management before and after participating in the mobile lung function testing service. The introduction of validated disease-specific questionnaires would assist in determining those interventions and/or models of care as associated with important improvements in the diagnosis and management of chronic lung disease.



There were several limitations of this study, including the small sample size, although a majority of individuals who had a lung function test agreed to participate in the study. This may have limited the ability to identify significant differences between rural and metropolitan participants. The wide range of the ACT scores from 5 to 25 also limits such comparisons. As the study recruited participants from only two metropolitan and two rural clinics, it is not possible to generalise these findings to the rest of WA, or more broadly to Australia. The research team relied on accurate and honest responses from participants about their respiratory conditions and medications prescribed. It was beyond the scope of the study to have access to the lung function test results. To obtain such data would require the permission of patients and GPs and we are unable to, under ethical considerations, access information on specific patients or GPs. A further study is needed to include this information.

The poor response rate from GPs was disappointing, especially as many were unaware of the lung function testing service. There should be greater awareness of the mobile lung function testing service, especially in rural areas of WA, not only for GPs but also for patients, as patients are able to self-refer to the service.

## 6. Conclusions

A majority of participants who sought respiratory function testing had poorly controlled asthma although their perception of asthma control did not correlate with the ACT score. Possible toleration of asthma symptoms may hinder GP visits. Participants with COPD and worsening of symptoms at follow-up may be partly explained by continued exposure to a smoky or dusty environments or seasonal variation. As there were high satisfaction rates with the service, strategies should be implemented to increase awareness of the mobile lung function-testing service. A further study to investigate the impact of the service on a larger scale is required.

**Author Contributions:** P.C., B.S. and K.W. developed the study concept and methodology, with contributions from G.L.H., P.C., B.S. and K.W. in developing the questionnaires. P.C. was responsible for funding acquisition, managed the overall project, sought ethics approval as well as approval to use validated tools. P.C., K.W. and F.F. recruited participants. R.P. completed data analysis. P.C., B.S. and R.P. prepared the original draft manuscript, which was reviewed and edited by P.C., B.S., R.P. and F.F. All authors have read and agreed to the published version of the manuscript.

**Funding:** This research was funded as part of the Curtin University School of Pharmacy and Biomedical Sciences Early to Mid-Career Academic Support Program.

**Institutional Review Board Statement:** This study was approved by the Curtin University Human Research Ethics Committee (HREC) (HRE2020-0130) on the 19 March 2020.

**Informed Consent Statement:** Informed consent was obtained from all subjects involved in the study.

**Data Availability Statement:** Data are available on request.

**Acknowledgments:** The authors would like to thank Bill Smith, respiratory scientist and CEO of Respiratory Testing Services for providing access to clinics. We would all like to thank the respiratory scientists at each clinic involved with this study, for their support.

**Conflicts of Interest:** The authors declare no conflict of interest.

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
