# Peer review of "A Preliminary Investigation of Mobile Respiratory Function Testing in Western Australian Communities"

_applsci, doi:10.3390/app13031855_

Round 1
Reviewer 1 Report
This paper investigates the potential benefits of mobile respiratory testing in Western Australia.
The authors clearly present the potential of pulmonary testing for Asthma and COPD management, and introduce the need of increasing accessibility to these tests.
By means of surveys, the authors collected the perceived satisfaction and experience with the mobile testing service, both from patients and from GPs.
I consider that the questionnaires were well designed, with most of the fields being associated to a scalar number or yes/no answers. Moreover, data from the questionnaires were acquired in digital form by design via an iPad (baseline), while follow up questionnaires were sent by email. I consider that this minimizes mistakes while maximizing patient adherence especially in the follow up. Data were stored in RedCap and analyzed with standard statistical software (SAS).
My main suggestions and concerns are the following.
1. Please explain better what you mean by "mobile" already in the abstract. It may not be obvious to readers of the journal with different background.
2. By providing two questionnaires to patients (baseline and follow up), the authors collected also data on the prevalence and proression of the disease, which are beyond the evaluation of satisfaction and experience with the mobile testing service. In my opinion these data are important and I would stress in the discussion that the availability of disease-specific questionnaires at tertiery centers or clinics may be useful to gather these types of data.
3. In the discussion the authors declare to have not access to the results of the tests perfomred by the patients nor to GP reports. Despite this does not compromise the paper itself, it may hamper futher exploitations of the results. Hence, if possible, I suggest adding a statement on how to obtain these data if some reader is interested in evaluating them.
4. I strongly encourage the authors to deposit the anonymized answers to the questionnaires in an Open Research Data repository, or as supplementary material to the current paper. These may be useful to other readers for further analyses.
In conclusion, I support the publishing of this paper with minor revision.
Author Response
This paper investigates the potential benefits of mobile respiratory testing in Western Australia.
The authors clearly present the potential of pulmonary testing for Asthma and COPD management, and introduce the need of increasing accessibility to these tests.
By means of surveys, the authors collected the perceived satisfaction and experience with the mobile testing service, both from patients and from GPs.
I consider that the questionnaires were well designed, with most of the fields being associated to a scalar number or yes/no answers. Moreover, data from the questionnaires were acquired in digital form by design via an iPad (baseline), while follow up questionnaires were sent by email. I consider that this minimizes mistakes while maximizing patient adherence especially in the follow up. Data were stored in RedCap and analyzed with standard statistical software (SAS).
Thank you for your comments.
My main suggestions and concerns are the following.
- Please explain better what you mean by "mobile" already in the abstract. It may not be obvious to readers of the journal with different background.
We have clarified ‘mobile’ in the abstract and introduction as follows:
Aim:
The aim was to investigate a mobile (ie transportable) respiratory testing service………..
Introduction:
Alternative testing is via private respiratory laboratories, some of which provide a mobile (ie transportable) lung function testing service
- By providing two questionnaires to patients (baseline and follow up), the authors collected also data on the prevalence and proression of the disease, which are beyond the evaluation of satisfaction and experience with the mobile testing service. In my opinion these data are important and I would stress in the discussion that the availability of disease-specific questionnaires at tertiery centers or clinics may be useful to gather these types of data.
The following has been added to the Discussion:
This study provided two questionnaires to participants (baseline and follow-up) to investigate patient disease management before and after participating in the mobile lung function testing service. The introduction of validated disease-specific questionnaires would assist in determining those interventions and/or models of care as associated with important improvements in the diagnosis and management of chronic lung disease
- In the discussion the authors declare to have not access to the results of the tests perfomred by the patients nor to GP reports. Despite this does not compromise the paper itself, it may hamper futher exploitations of the results. Hence, if possible, I suggest adding a statement on how to obtain these data if some reader is interested in evaluating them.
We have added the following to the discussion:
It was beyond the scope of the study to have access to the lung function test results. To obtain such data would require the permission of patients and GPs and we are unable to, under ethics, access information on specific patients or GPs. A further study is needed to include this information.
- I strongly encourage the authors to deposit the anonymized answers to the questionnaires in an Open Research Data repository, or as supplementary material to the current paper. These may be useful to other readers for further analyses.
This was a pilot study to assist in future study designs. The results from this study will be used to inform further development of mobile respiratory testing services within the context of primary healthcare in Western Australia. Readers with an interest in the anonymised dataset can contact the corresponding author.
In conclusion, I support the publishing of this paper with minor revision.
Thank you for your support

Reviewer 2 Report
This work is concerned with the preliminary investigation of mobile respiratory function testing in Western Australian communities. Minor mistakes required correction, as follows:
1. Please review the English and grammar in the article.
2. Provide the full form of abbreviations at the time of first use, thereafter use the abbreviation, check the article.
Author Response
Reviewer 2
This work is concerned with the preliminary investigation of mobile respiratory function testing in Western Australian communities. Minor mistakes required correction, as follows:
Please review the English and grammar in the article
The English and grammar has been addressed throughout the manuscript.
Provide the full form of abbreviations at the time of first use, thereafter use the abbreviation, check the article.
This has been corrected to ensure all abbreviations have been defined at the first time of use and thereafter the abbreviation was used

Reviewer 3 Report
The manuscript investigates a mobile respiratory testing service in 2 metropolitan and 2 rural areas of Western Australia based on 59 individuals. Survey results are shown in tables and compared in different areas or between baseline and follow-up scores. I think the study is meaningful but still have questions about its results. My 10 comments were attached.

Author Response
Reviewer 3
The manuscript investigates a mobile respiratory testing service in 2 metropolitan and 2 rural areas of Western Australia based on 59 individuals. Survey results are shown in tables and compared in different areas or between baseline and follow-up scores. I think the study is meaningful but still have questions about its results. My 10 comments were attached.
Thank you for your comments
- The smoking status include never, current, and former smokers in Table What are the definitions or survey questions for different status?
Although the different statuses were not formally defined for respondents, the terms were understood to mean:
- Never – the respondent had never smoked a cigarette.
- Current – the respondent is currently a smoker.
- Former – the respondent had previously been a smoker but had quit.
Respondents who indicated they were a current or former smoker were also asked whether they were considering quitting smoking or how long ago they quit, respectively.
- In table 1, if the sum of sample sizes n is not 59 please explain the reason, e.g. participants can choose more than one answer or the question was conditionally asked certain people.
In Table 1, the data shows n=59 for many parameters. However, for the parameter ‘reasons for lung function test’ and ‘medical conditions’ respondents were asked to tick all responses that apply.
Respondents were asked to tick one box for ‘diagnosis of asthma’ and ‘diagnosis of COPD’ but two respondents ticked more than one box for ‘diagnosis of COPD’ (one respondent ticked doctor + medical specialist; another respondent ticked doctor + medical specialist + during a hospital admission).
For each parameter, the ‘n’ value has been included in Table 1 and where respondents were asked to tick all that apply, this has been specified.
- What’s the meaning of “>10” after “Most common medical conditions” in Table 1? If 10 is the number of medical conditions, why are there only 7 groups (Anxiety…High cholesterol) in Table 1.
In Table 1 for ‘most common medical conditions’, >10 means that greater than 10 respondents indicated they had that specific medical condition. This has been clarified in Table 1 in the paper.
- In Table 1, 8 people reported that they had “Asthma Action Plan” (vs. 22 who had no plan), however, the corresponding proportion is 3% (vs. 26.7%). The proportions are wrong.
Thank you. This has been corrected.
- In the first paragraph of the section “Respondents with asthma or COPD”, “although one respondent (3.3%), who lived remotely in Narrogin, had to go to hospital 12 times in the past 12 months for these reasons.” I cannot find the result in any table or figure. What happened for the rest 5 participants (30-24-1=5)?
The following narrative has been added to the results:
Most (24/30; 80.0%) did not have to go to hospital or the Emergency Department in the past 12 months due to worsening or out of control asthma symptoms, although four respondents (13.3%) had a single hospital presentation in the past 12 months, one respondent (3.3%) had two presentations, and another respondent (3.3%), who lived remotely in Narrogin, had 12 hospital presentations in the past 12 months for these reasons
- In the second paragraph of the section “Respondents with asthma or COPD”, “three had referred to it in the past 12 months for the management of their asthma. Most (7/8; 87.5%) felt confident to use their Asthma Action ” I cannot find these results in any result table.
These results were just presented in narrative form as a statement, as a Table was considered unnecessary.
- In the third paragraph of the section “Respondents with asthma or COPD”, “Overall 11/36 (30.5%) were potentially incorrectly ” How did you get 11 and 36? Are these numbers in baseline or follow-up surveys? What’s the definition of “potentially incorrectly diagnosed”? I cannot get these numbers from the result tables.
There were 5 respondents who identified having asthma at baseline although following the lung function test, they were not diagnosed with asthma. Hence, the fact they reported they had asthma at baseline was ‘potentially incorrectly diagnosed’.
Following the lung function test, 6 respondents who did not have asthma at baseline, were identified as having asthma. Without undergoing lung function testing, these individuals were also ‘potentially incorrectly diagnosed’. Hence 11 participants (5 + 6) were ‘potentially incorrectly diagnosed’.
At baseline, 30 respondents reported they had asthma. Following lung function testing, an additional 6 respondents reported a diagnosis of asthma (hence the denominator is 30+6 ie 36).
- In Table 4, how did you get the values for “change” between baseline and follow-up? For example, the baseline ACT score is 16.12 and the follow-up score is 16.88 in Metropolitan, however, the change is 0.43 rather than 0.76 (=16.88-16.12). Please add more explanation.
The reason that the baseline score cannot be subtracted directly from the follow-up score is that the denominator for the baseline results may not be the same as the denominator for the follow-up results. The means for baseline and follow-up are based on all the available data, but for the change we could only use records for individuals who completed both baseline and follow-up surveys.
- In Table 4, there is a “*” for the p-value of the follow-up CCQ total score. Does that mean only this value is the Satterwaite p-value and other p-values in the last column using different methods? If yes, please explain why you use different methods; if not, please change the position of “*” to avoid misunderstandings.
Yes, this is correct. The values provided in Table 4 were provided by statistician, Dr Richard Parsons who calculated p-values obtained from two t-tests (t-test and t-test Satterwaite), comparing the various scores between metro and rural. These cohorts may not have the same variances. Further, the F-test was used to indicate whether the p-values from the t-test and t-test Satterwaite should be used, based on whether the F-test was significant (p<0.05). There was only one result where this occurred, hence the Satterwaite p-value was reported. For all other results in Table 4, the standard t-test p-values were reported.
For clarity, the following has been added under the heading ‘Analysis’:
Differences in various validated test scores at baseline and follow-up between metropolitan and rural areas were determined. p-values were obtained from two t-tests (t-test and t-test Satterwaite), comparing the various scores between metro and rural, as these cohorts may not have the same variances. Further, the F-test was used to indicate whether the p-values from the t-test and t-test Satterwaite should be used, based on whether the F-test was significant (p<0.05). Where this occurred, the Satterwaite p-value was reported. For all other results, the standard t-test p-values were reported
- The first two sentences in the discussion section are “Most participants had poorly controlled asthma indicated by an ACT <=19. Despite this, many perceived their asthma to be well-controlled.” I can understand the first sentence based on 73.3% of individuals having <=19 ACT However, how did you get the second conclusion based on the survey results?
We have changed the second sentence in the discussion as shown below:
Most participants had poorly controlled asthma indicated by an ACT £19. Despite this, six respondents who had an ACT £19 perceived their asthma to be well-controlled.

Reviewer 4 Report
I read the the manuscript with high interest. I have the following comments that should be addressed:
1. A different references style is observed in the Introduction:
Although many General Practitioner (GP) surgeries have spirometers, use of spirometry testing is low.9, 10
2.
Table 3. Comparison of validated test scores at baseline and follow-up.
Why the authors did not statistically compare between the paired participants?
3. Why the authors in Table 4 used the uncommon 5 decimal points for p values?
4.Start the discussion part with a clear aim of the study
5. In the discussion, where are the statistics that support the following sentence: An unexpected finding in this study was that the ACT score did not improve significantly after follow-up, when it was expected participants would have consulted their GP to optimize therapy.
6. More improvement of the discussion part is recommended
Author Response
I read the the manuscript with high interest. I have the following comments that should be addressed:
- A different references style is observed in the Introduction:
Although many General Practitioner (GP) surgeries have spirometers, use of spirometry testing is low.9, 10
Thank you for your comment. We have corrected this.
- Table 3. Comparison of validated test scores at baseline and follow-up.
Why the authors did not statistically compare between the paired participants?
We have added p-values into Table 3, for the comparison for each variable from baseline to follow-up. While there are statistically significant changes for some variables, the sample sizes for most variables are small.
We have added the following to the manuscript under results:
In comparing each variable from baseline to follow-up, while there are statistically significant changes for some variables, the sample size for most variables are small (Table 3).
- Why the authors in Table 4 used the uncommon 5 decimal points for p values?
We have reduced the p values in Table 4 to 3 decimal places.
- Start the discussion part with a clear aim of the study
We have modified the discussion as follows:
The aim of this pilot study was to investigate the potential benefits and overall impact of a mobile respiratory function testing service in metropolitan and rural areas of WA. We found that the majority of participants were satisfied with the lung function testing experience and would recommend the service to others. However, most GPs were not aware of the mobile lung function testing clinic.
- In the discussion, where are the statistics that support the following sentence: An unexpected finding in this study was that the ACT score did not improve significantly after follow-up, when it was expected participants would have consulted their GP to optimize therapy.
We have included p-values for paired t-tests. For the ACT score, there is no significant difference between baseline and follow-up scores,
We have modified the discussion to:
A finding in this study was that the ACT score did not improve significantly after follow-up, when it was expected participants would have consulted their GP to optimise therapy. Notably eight weeks after the lung function test, some participants had not consulted their GP, which may have been due to a lack of appointment availability. However, given the lack of GP awareness of the mobile lung function testing clinic, found in this pilot study, and not knowing when the respiratory physician’s report was sent to the GP, there is very little chance that the service can improve disease outcomes.
- More improvement of the discussion part is recommended
The Discussion has been reconsidered and considerable changes have been made to improve it.
